

# The influence of fundamental frequency on perceived duration in spectrally comparable sounds

Caitlin Dawson[1], Daniel Aalto[2,3], Juraj Simko[4] and Martti Vainio[4]

[1] Department of Psychology and Logopedics, University of Helsinki, Helsinki, Finland
[2] Communication Sciences and Disorders, Faculty of Rehabilitative Medicine, University of Alberta, Edmonton, Canada
[3] Institute for Reconstructive Sciences in Medicine, Misericordia Community Hospital, University of Alberta, Edmonton, Canada
[4] Department of Modern Languages, University of Helsinki, Helsinki, Finland

## ABSTRACT

The perceived duration of a sound is affected by its fundamental frequency and intensity: higher sounds are judged to be longer, as are sounds with greater intensity. Since increasing intensity lengthens the perceived duration of the auditory object, and increasing the fundamental frequency increases the sound's perceived loudness (up to ca. 3 kHz), frequency modulation of duration could be potentially explained by a confounding effect where the primary cause of the modulation would be variations in intensity. Here, a series of experiments are described that were designed to disentangle the contributions of fundamental frequency, intensity, and duration to perceived loudness and duration. In two forced-choice tasks, participants judged duration and intensity differences between two sounds varying simultaneously in intensity, fundamental frequency, fundamental frequency gliding range, and duration. The results suggest that fundamental frequency and intensity each have an impact on duration judgments, while frequency gliding range did not influence the present results. We also demonstrate that the modulation of perceived duration by sound fundamental frequency cannot be fully explained by the confounding relationship between frequency and intensity.

## INTRODUCTION

A simple sinusoid sound is characterized by three parameters: duration, frequency, and intensity. The subjective and objective duration of a sound (or other sensory stimulus) are not equal (*Eagleman & Pariyadath, 2009*; *Eisler, 1976*) and depend on the state of the observer (*Cheng, MacDonald & Meck, 2006*) and the organization of consecutive sounds in time (*Lake, LaBar & Meck, 2014*), as well as the structure of the sound signal itself. In particular, sound signals with larger amplitudes are perceived as longer when compared with sounds of the same objective duration but smaller amplitude, indicating that sound intensity contributes to duration perception (*Berglund et al., 1969*). The perceived duration

Corresponding author
Caitlin Dawson,
caitlin.dawson@helsinki.fi

of sounds is also prolonged by frequency, with higher tones being perceived as longer than lower ones; this has been documented for simple sinusoid sounds (*Burghardt, 1972*) as well as for more spectrally complex tones (*Lehiste, 1976*; *Rosen, 1977*; *Jeon & Fricke, 1997*). Analogous modulation phenomena exist in visual and tactile sensory modalities (*Goldstone, Lhamon & Sechzer, 1978*; *Kraemer, Brown & Randall, 1995*; *Rammsayer & Verner, 2014*; *Ekman et al., 1969*).

However, it is not clear whether intensity and fundamental frequency can influence the perceived duration of sounds independently, and to what extent. Since the frequency of sinusoidal sounds contributes to their intensity as well as to their perceived loudness (*Moore, 2012*), the modulation effect of frequency on perceived duration could potentially be explained as a byproduct of the modulation of perceived duration by intensity alone.

The duration coding mechanism that might be partly responsible for the observed frequency modulations in duration perception could be based on two different phenomena: first, the neural spikes corresponding to the onset or offset of the sound may be delayed as a function of intensity and phase-locked modulation frequency (i.e., fundamental frequency) (*Jeon & Fricke, 1997*) (see also *Joris, Schreiner & Rees, 2004* for neurophysiological evidence); second, the accumulation of input spikes may be affected by the intensity and frequency of the sound (as in the dual klepsydra model *Wittmann, 2013*).

In addition to low level mechanisms, several high level explanations for the duration modulation by frequency have been proposed. According to a Gestalt approach, a high frequency sound is perceived as intrinsically smaller—and consequently, shorter—than a low frequency sound (*Brigner, 1988*). Also, speech-like auditory objects may be subject to a compensatory strategy as high pitched syllables in many languages (e.g., lexical tones in Mandarin Chinese) are produced shorter than low-pitch tones (*Gussenhoven & Zhou, 2013*). (This explanation can, of course, be reversed, with pitch-dependency of the production patterns interpreted as a compensatory consequence of perceptually driven pitch-modulation of perceived duration, e.g., *Yu, 2010*).

Disentangling influences of intensity and fundamental frequency on perceived duration of a sound is demanding for several reasons. First and foremost is the fact that intensity and frequency are themselves not independent from each other. Physically, intensity of a (simple) sound is a function of its frequency. As equal loudness contours show, the sound frequency also positively contributes to its perceived loudness up to about 3.2 kHz.

Apart from purely psychoacoustic considerations, the interactions between fundamental frequency, intensity and duration perception are also of linguistic and phonetic interest. Relative prominence of speech constituents, encoding lexical stress, emphasis, etc., is signalled through a complex, context-sensitive and language-dependent interplay between these three sound characteristics (as well as spectral properties), which underline the necessity of stimulus choice in experimental settings. While sinusoidal sounds are an important research tool, they are scarce in the biologically plausible sound environment of human communication where sounds are complex and the fundamental frequency is constantly moving. In tone languages, syllable duration cannot be separated from the underlying tone (*Gandour, 1977*; *Kong, 1987*; *Gussenhoven & Zhou, 2013*). In quantity languages, the fundamental frequency ($f_o$) contours contribute to distinguishing between
quantity categories (*Suomi, 2005*; *Vainio et al., 2010*; *Lippus et al., 2013*). In addition to $f_o$ level, its movement also interacts with perception of syllable duration, albeit in somewhat language-dependent way (*Lehiste, 1976*; *Rosen, 1977*; *Gussenhoven & Zhou, 2013*; *Lehnert-LeHouillier, 2010*; *Šimko et al., 2015*; *Cumming, 2011*); this interaction may depend on the linguistic or non-linguistic goals of the experimental paradigm.

In the present work, we revisit this issue with a special focus on disentangling the contributions of fundamental frequency and intensity to the perceived duration of a sound. We report the results of two forced choice experiments in which the listeners were asked to compare the durations and intensity levels, respectively, of two sounds varying simultaneously in multiple dimensions of duration, fundamental frequency and intensity. The possible confounds between fundamental frequency and intensity are addressed in two ways. First, the sounds presented to the participants have rich spectral content that is band pass filtered to a narrow spectral frequency range well separated from the fundamental frequency. This results in a pitch sensation that corresponds to the fundamental frequency although almost no energy is left at that frequency. This percept is sometimes named the missing fundamental phenomenon since the auditory nerve fibres corresponding to the fundamental are not encoding the signal. Instead, the amplitude modulated movement of the basilar membrane at the band pass frequencies conveys the information with slow oscillations corresponding to the fundamental. Second, using logistic linear regression analysis, the effects of fundamental frequency on both duration and intensity judgments are quantified. Comparing these influences allows us to evaluate whether the contribution of fundamental frequency to loudness is sufficient to explain its effect on duration perception.

## METHODS

### Participants

In a pilot phase and in an earlier work where similar stimuli were used, the fundamental frequency had an impact on duration judgments of each participant (*Šimko et al., 2015*). Since every individual recruited for the study was expected to show fundamental frequency modulation of the duration judgment on an individual level, there was no constraint to the minimum sample size. If a participant was not affected by fundamental frequency in their duration judgments, they would be excluded from the analyses since the experimental question for these individuals would not make sense.

Eleven native monolingual Finnish speakers aged 18–40 participated in the experiment. They were screened for normal hearing (<20 dB). The stimuli were presented through headphones that were calibrated for each participant so that the standard sound always had a fixed intensity level of 66 dB SPL (A-weighted) measured by a PeakTech 5055 sound level meter. Every participant took part in both discrimination tasks reported here.[1] The experiments were performed according to the guidelines of the Declaration of Helsinki at the University of Helsinki, Finland; the Committee for ethical review granted ethical approval to carry out the study within its facilities and the participants gave their written consent to take part in the experiments.

[1] In addition, the participants also performed two additional experiments preceding the discrimination tasks reported here.

## Procedure

Two 2-alternative forced choice discrimination tests were performed to evaluate multifeature intensity and duration discrimination, respectively. The sounds presented to the participants were varied in four parameters: duration, $f_o$ level, dynamic $f_o$ range, and intensity level, and were drawn at random so that the probability distribution of each parameter formed a truncated normal distribution.

The tasks were performed in two blocks (intensity discrimination block and duration discrimination block) and the order of the blocks was randomized between the subjects. In each block, the participants were presented with 300 pairs of stimuli. After presentation of each pair they were asked to identify which of the two sounds was louder or longer, respectively, by typing $a$ (for the first) or $x$ (for the second) on a standard (Finnish) QWERTY keyboard.

## Signal generation

Every sound stimulus was fully characterized by four parameters—duration, $f_o$ level, dynamic $f_o$ interval and intensity level—drawn for each sound from a truncated normal probability distribution (if a randomly generated parameter was more than two standard deviations away from the mean, it was discarded and a new value was generated). The duration of each sound was drawn independently from the normal distribution with mean 300 ms and standard deviation 75 ms. $f_o$ level followed a truncated normal distribution with mean 150 Hz (corresponding to 0 semitones) and a standard deviation of 4 semitones (hence, there were more sounds over 200 Hz than under 100 Hz). Here the semitone is defined as the twelfth root of two, i.e., the fixed ratio between frequencies of adjacent keys of an equally tempered keyboard. The dynamic $f_o$ range followed a truncated normal distribution with mean 0 (static sound) and standard deviation of 4 semitones, and the intensity level followed a truncated normal distribution with mean 66 dB and standard deviation 3 dB. Altogether, the durations varied from 150 ms to 450 ms and the instantaneous pitch of the sound varied between 75 Hz and 300 Hz (this is 8+4 semitones below and above 150 Hz). Since the gammatone filter used for signal generation (see below) had a center frequency over 3 kHz, even the highest instantaneous $f_o$ had only unresolved harmonics within the band. At the biomechanical level of the cochlea, the basilar membrane vibrations are amplitude modulated to the fundamental frequency, creating the same pattern in the auditory nerve signal. With sufficient separation (as here) between the active spectral band and the fundamental frequency, the individual harmonics of the sound should not be discernible. The intensity levels varied between 60 dB and 72 dB. Because the distributions are truncated, the true standard deviations of the generated distributions are slightly smaller.

For each randomly generated set of parameters, a positive sawtooth wave of the given duration was frequency modulated so that the instantaneous fundamental frequency was exponentially increasing/decreasing depending on the sign of the interval, and the frequency at the middle of the signal duration corresponded to the given $f_o$ level (see Fig. 1A). To avoid a jump in the waveform at the end of the signal, the predetermined

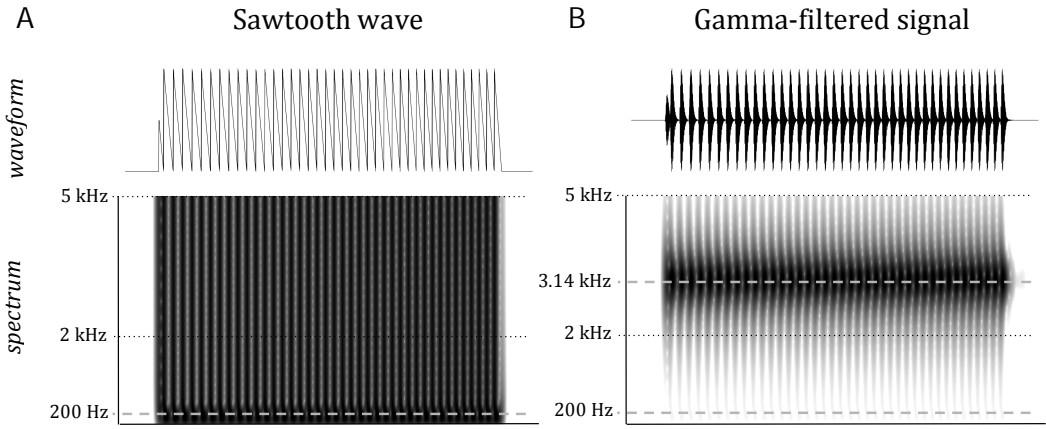

**Figure 1** **The waveform and spectrum of the original (A) and gamma-filtered (B) sawtooth wave.** The wave mid-point fundamental frequency is 200 Hz, duration 200 ms. The frequency movement interval is 4 semitones. Note that the energy in the filtered signal is centered around the gammatone filter's center frequency of 3.14 kH, and is negligible around the signal's fundamental frequency of 200 Hz.

duration was prolonged to the next zero crossing. The sawtooth waveform was selected to ensure a rich spectral content on the targeted frequency band.

The sawtooth wave was subsequently band-pass filtered using a 4th order approximation of the gammatone filter with center frequency $f_c = 3141.6$ Hz (*Cooke, 1993*) (Fig. 1B). The center frequency was chosen so that the fundamental frequency was always well separated from the frequency band containing the signal energy. Further, a normalization of the energy of the signal was performed to even out residual loudness differences; the normalization was based on the energy average over the first 100 ms of the waveform. Finally, the signal amplitude was changed to reach the desired intensity level, completing the single sound generation.

Having generated the pair of stimuli for each trial in this way, the sounds were joined using randomly generated inter-stimulus interval: a duration was drawn from a truncated normal distribution with mean 800 ms and standard deviation 10 ms, and used as an interval from the onsets of the first to the onset of the second sound. Then, 600 ms of silence was added before the onset of the first sound and after the offset of the second sound and a white noise was added (to the entire signal, including these pre- and post-stimulus intervals) with 10 dB signal to noise ratio in order to mask nonlinear distortion tones and guarantee a narrow spectral band. In the pilot phase, the stimuli presented without noise did not have a perceivable distortion product but for very similar sounds presence or absence of noise has reportedly changed their pitch percept (*Pressnitzer & Patterson, 2001*). Finally, linear onset and offset ramps covering the first and the last 200 ms of the entire stimulus containing the two sounds were created. Figure 2 shows an example of one trial.

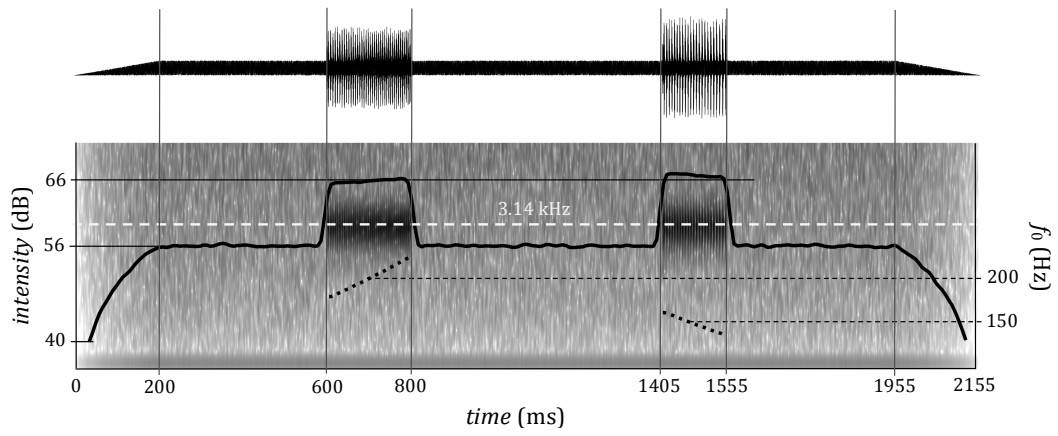

**Figure 2** An example of a trial with two stimuli with waveform (top) and spectrum (bottom). The fundamental frequency and duration parameters of the first stimulus are identical with that in Fig. 1, the second stimulus is 150 ms long, with central frequency of 150 Hz and $f_o$ movement interval −3 semitones. The gain for the first signal is 0 dB, for the second 0.5 dB. Inter-stimulus interval of this example trial (between the onsets of the sounds) is 805 ms. The central frequency of the gammatone filter is superimposed over the spectrogram.

## Predictability and correlations between the design factors

Because the final periods of the sounds were completed, duration and fundamental frequency were somewhat correlated. In order to evaluate the amount of resulting interaction between the variables and its impact on the actual durations of the stimuli, linear regression models were fitted to the actual acoustic characteristics of the stimuli that were played to the participants.

A third order non-linear regression model with the sound duration as a dependent variable and $f_o$ level and dynamic $f_o$ interval as predictors explained 0.02% of the durational variation with a significant effect of the $f_o$ level; the dynamic $f_o$ interval effect was not significant. The small artificial lengthening of the sounds thus does not provide strong cues to the stimulus duration. In fact, the amount of durational manipulation was typically small, on average 1.5 ms.

The $f_o$ gliding speed, on the other hand, gives much more information. A linear model with gliding speed as an independent and duration as a dependent variable fitted to the generated stimuli explained 9.5% of the variation and a 15th order nonlinear model explained 11.6% of the variation (increasing the order of the model beyond 15 did not increase the $r^2$ value any more). The effect size as estimated from the (first order) linear model is quite large: the statistical model predicts a 23 ms shorter duration for a sound with a change rate of 10 semitones per second as opposed to a static sound of the same duration. This is the largest single source of bias in the data; however, it should not affect the results since participants would not know that the gliding speed is related to duration and they should not be able to detect it during the experiment.

In fact, the design of the current work was such that these biases work, in general, against the hypotheses that higher and more dynamic $f_o$ are perceived as longer. As the last period of each generated sound was prolonged up to the end of the last period, the low $f_o$ stimuli and falling sounds were effectively lengthened more than the high/rising ones. Additionally, in the current design, the probability of a long sound was reduced by observing a very fast moving stimulus—if a subject inferred this relationship, she or he would consequently judge the more dynamical stimuli as shorter rather than longer.

## Statistical analysis of the response data

Separate mixed effect logistic regression (logit) models were fitted for each discrimination task, with participants' response as a dependent variable (with values 1: the first sound louder/longer, and 0: the second sound louder/longer). The fixed effects were the differences between the actual signal parameters of the two stimuli in each trial (a value for the first sound minus the corresponding value for the second sound): the difference between durations (in s), the difference between intensity levels (in dB), the difference between $f_o$ levels (in semitones), and the difference between the dynamic $f_o$ ranges.

For the last difference variable we tested two alternative versions, the first representing the difference in absolute dynamicity (the difference between *absolute* values of the $f_o$ ranges), and the second representing the difference in dynamicity including direction of the fundamental frequency slope (the difference between the *raw* values of the $f_o$ ranges). All the manipulated variables (differences) were used as random slopes for the subjects.

Full models with interactions among these fixed effects as well as models without interactions were fitted. Statistical significance was estimated using Monte Carlo simulations as implemented in the R package lme4 (*Bates et al., 2015*). Deviance reduction was calculated for the individual models in order to assess the quality of fits (*Pinheiro & Bates, 2000*).

In addition, in order to compare the performance of individual participants with the group effects (captured by the mixed effect models), we fitted simple logistic models for the same dependent and fixed effect variables as above (without interactions) for each participant separately.

## RESULTS

### Intensity discrimination

Logistic regression analysis of the data shows that duration difference, $f_o$ difference, and intensity difference all had a significant impact on intensity judgments. The dynamic $f_o$ range difference did not reach significance, whether conceived as a difference in absolute (Table 1) or raw (Table 2) values of the range. The interactions were not significant, therefore only the coefficients from the models containing just the main effects are reported.

The effects of duration, intensity, and fundamental frequency differences are all positive and highly significant, indicating that increases in any of these parameters are associated with a perceived increase in loudness (the greater the difference, the relatively greater the parameter value for the first sound, and, according to the model, the greater likelihood of a 'first sound louder' response). Also relevant are the almost identical values of the estimates

**Table 1** Mixed effects model fitted to the responses of *intensity discrimination* with frequency range difference calculated as the difference between the absolute values of the dynamic $f_o$ ranges.

| Effect | Size | Error | z value | p (MCMC) |
|---|---|---|---|---|
| Intercept | 0.14 | 0.076 | 1.8 | 0.07 |
| Duration difference | 6.5 | 0.71 | 9.2 | $2 \cdot 10^{-16}$ |
| Intensity difference | 0.34 | 0.055 | 6.2 | $4 \cdot 10^{-10}$ |
| Frequency difference | 0.14 | 0.036 | 3.9 | $1 \cdot 10^{-4}$ |
| Frequency range difference | 0.028 | 0.020 | 1.4 | 0.2 |

**Table 2** Mixed effects model fitted to the responses of *intensity discrimination* with frequency range difference calculated as the difference between the raw values of the dynamic $f_o$ ranges.

| Effect | Size | Error | z value | p (MCMC) |
|---|---|---|---|---|
| Intercept | 0.14 | 0.0768 | 1.8 | 0.07 |
| Duration difference | 6.6 | 0.68 | 9.6 | $2 \cdot 10^{-16}$ |
| Intensity difference | 0.34 | 0.054 | 6.3 | $2 \cdot 10^{-10}$ |
| Frequency difference | 0.14 | 0.036 | 3.9 | $1 \cdot 10^{-4}$ |
| Raw frequency range difference | $-0.006$ | 0.011 | $-0.57$ | 0.56 |

of these fixed effects between the two models with different dynamic range comparison variables.

By comparing deviance of the experimental models to the deviance of the null model which includes only random intercepts for the subjects, the current models both reduce approximately 26% of the deviance for intensity judgments, showing that the addition of the modulating variables produces a better fit to the data for both models.

Comparing the model parameters allows us to compare the relative effects of sound manipulation between different acoustic parameters. Increasing the intensity level of the first sound by 1 dB adds 0.34 to the linear combination of the dependent variables in the inverse logit function fitted by the model. The same quantitative effect can be achieved by 52 ms (0.052 s) durational lengthening of the sound ($6.5 \times 0.052 \simeq 6.6 \times 0.052 \simeq 0.34$) or 2.4 semitones increase in fundamental frequency ($0.14 \times 2.4 \simeq 0.34$).

The standard deviations of the random slopes were 1.8 and 1.6 for duration, 0.18 and 0.17 for intensity, 0.11 and 0.11 for fundamental frequency, and 0.05 and 0.02 for frequency range (respectively, for the two models). Comparing these estimates to the effect sizes indicate more inter-subject variability in frequency range response compared to the other signal parameters (for this variable only, the standard deviation is actually greater than the estimate).

This assessment is explicitly confirmed by the separate logistic models with the same dependent and independent variables fitted for each participant individually (the estimates and their significances are reported in full in Tables A1 and A2). The duration difference and intensity difference variables are significant and positive for both versions of the model and for all eleven participants. Also, the effect of $f_o$ difference is significantly positive for

**Table 3   Mixed effects model fitted to the responses of *duration discrimination* with frequency range difference calculated as the difference between the absolute values of the dynamic $f_o$ ranges.**

| Effect | Size | Error | z value | p (MCMC) |
|--------|------|-------|---------|----------|
| Intercept | 0.47 | 0.19 | 2.4 | 0.016 |
| Duration difference | 29 | 2.8 | 10 | $2 \cdot 10^{-16}$ |
| Intensity difference | 0.073 | 0.018 | 4.1 | $4 \cdot 10^{-5}$ |
| Frequency difference | 0.19 | 0.029 | 6.8 | $1 \cdot 10^{-11}$ |
| Frequency range difference | 0.021 | 0.020 | 1.0 | 0.3 |

all but two subjects (subject number 1 and 8) in both models. On the other hand, the dynamicity difference is only significant (and positive) for one participant in the case of difference in absolute range and two participants (one negative, the other positive) for the directional difference in raw dynamic ranges.

## Duration discrimination

Duration difference, intensity difference, and $f_o$ difference had a significant effect on duration discrimination, but dynamic $f_o$ range difference did not reach significance (Table 3). The interactions were not significant; therefore only the coefficients from the models containing just the main effects are reported.

Comparing deviance of the experimental models to the deviance of the null model which includes only random intercepts for the subjects, both current model reduce approximately 45% of the deviance for duration judgments, showing once again that the addition of the modulating variables produces a better fit to the data.

Using the same technique as in the previous section shows that the duration judgments were equally impacted by 10 ms duration increase, an intensity increase by 4.0 dB, and a 1.5 semitone fundamental frequency raise. An intensity increase by 1 dB corresponds to fundamental frequency increase of 0.38 semitone.

These findings hold for both models with the alternative definitions of the effect of dynamic $f_o$ range effects. However, for duration discrimination task, the directional effect of the $f_o$ dynamicity reached significance (Table 4) while the effect of difference in the absolute values of dynamic ranges did not (Table 3). The significantly positive sign of the raw frequency range difference estimates in Table 4 means that the stimuli with the rising $f_o$ (positive range value) tended to be judged as longer compared with the falling (or less rising) stimuli. (Reporting an equivalence between the range difference and the other difference variables, although straightforward to derive, would be very cumbersome and is therefore left out from this description).

The standard deviations of the random slopes were 8.5 and 8.8 for duration, 0.09 and 0.08 for frequency, 0.04 and 0.04 for intensity, and 0.03 and 0.03 for frequency range (respectively, for the two models). That is considerably smaller than the effect size for duration, and about half the estimate for frequency and intensity differences. This measure of inter-speaker variability was greater than estimate for the model using difference of the

**Table 4  Mixed effects model fitted to the responses of *duration discrimination* with frequency range difference calculated as the difference between the raw values of the dynamic $f_o$ ranges.**

| Effect | Size | Error | z value | p (MCMC) |
|---|---|---|---|---|
| Intercept | 0.45 | 0.19 | 2.4 | 0.018 |
| Duration difference | 29 | 2.9 | 10 | $2 \cdot 10^{-16}$ |
| Intensity difference | 0.072 | 0.018 | 4.0 | $8 \cdot 10^{-5}$ |
| Frequency difference | 0.19 | 0.029 | 6.9 | $5 \cdot 10^{-12}$ |
| Raw frequency range difference | 0.035 | 0.014 | 2.5 | 0.014 |

absolute range values and approximately the same as the estimate in the alternative model reported in Table 4.

The logistic regression models fitted for individual participants (see Tables A3 and A4) shed further light on the degree of robustness of effects regarding the participants' judgments. For both duration and $f_o$ level differences, the effect is consistently positively significant for all subjects. Somewhat surprisingly, the intensity difference effect is only significant for three participants out of 11 (subject numbers 5, 7 and 11 for both model types); however, the sign of the effect is also positive for all remaining participants even though the effect fails to reach significance. A similar situation arose for the model with raw range difference as the fixed effect (Table A4): although the estimates are positively significant for only two participants (number 3 and 11); in all other cases (except participants 1 and 2) the effect is also positive. On the other hand, for the model with difference in absolute range values (Table A3), the sign of the estimate is positive and negative for approximately half of the participants each, indicating a much greater degree of qualitative variability across our participants.

Furthermore, in a hierarchical series of models for duration judgment, a model including the intensity (but not frequency) term reduced 37.6% of the model deviance, whereas the model with the frequency (but not intensity) term reduced as much as 44.5% of the model deviance compared to 45.2% for the full model. This suggests that the impact of fundamental frequency on duration judgment is not entirely explained by the impact of intensity. Additionally, when interaction terms were added to the models, they were small and not significant.

## Comparing the discrimination results

The two tasks were based on identically generated stimuli allowing for the models for duration and intensity discrimination to be directly compared. While the leading term in both models corresponds to the primary acoustical correlate of perceived duration and loudness respectively, the duration and loudness judgments were both influenced by other sound parameters. Some (or all) of the fundamental frequency variation could lead to variations in perceived loudness that would then lead to duration modulation.

To interpret the results, a thought experiment is carried out where the impact of loudness, as opposed to the intensity, on duration judgments is estimated from the two experiments. Assuming that the impact of loudness (perceived intensity) on perceived

duration is proportional to the measured impact of intensity (since this is the leading term in the logistic regression model for loudness judgments), and assuming that a portion of fundamental frequency would impact the loudness which would then in turn impact the duration, the two logistic models can be combined by substituting the intensity term in the duration judgment model with a loudness term, which is a linear combination of duration, fundamental frequency, and intensity.

The quantitative estimate of fundamental frequency induced loudness effect does not support the idea that duration modulation by fundamental frequency would be primarily generated by intensity variations. In the loudness judgment phase, there was an equal impact of 1 dB and 2.4 semitones raise, whereas in the duration judgment phase, there was an equal impact of 1 dB and 0.38 semitones raise. To give an upper limit for fundamental frequency impact on the duration judgment through the loudness, not more than 0.38 semitones could be bundled to the loudness percept corresponding to 15.8% (0.38/2.4) of the total fundamental frequency effect.

## DISCUSSION

Clearly, duration perception is closely linked to both intensity and fundamental frequency. Higher fundamental frequency and greater intensity have been shown before to be associated with longer perceived duration. The presented set of experiments targeted the question of whether the impact of fundamental frequency on duration judgments is simply a confound, i.e., whether it can be explained through the influence of frequency on perceived intensity.

The results strongly suggest that this is not the case, and that fundamental frequency has an independent contribution to the perceived duration of a sound. In the light of the design presented here, this finding is supported on several levels. First, care was paid to neutralizing the interdependencies between intensity and fundamental frequency in our stimuli; nevertheless, participants' judgments in the duration discrimination task were significantly influenced by the stimulus frequency. Second, the quantitative comparison of duration and intensity judgments demonstrates that the fundamental frequency contribution to intensity judgments is not sufficient to fully account for its influence on perceived sound duration. Finally, as shown by the rather low deviance reduction by the models, the discrimination tasks were quite difficult for the participants. Still, they were able—in a statistical sense—to perform the discriminations, and their judgments were robustly influenced by most signal properties in expected directions (with the exception of $f_o$ dynamics).

Stimulus design parameters necessarily have an effect on the response patterns: larger variation in a primary acoustic correlate of a perceptual dimension (e.g., intensity level in loudness discrimination) makes the task easier. The differences between the two stimuli in a trial (determined by standard deviations of parameter distributions) must be detectable but not so large as to make the stimuli qualitatively incompatible with each other. Larger standard deviations in the complex stimuli would make a less difficult task and result in less guessing from participants, reducing the precision of the model estimates; however, a task that is entirely guessing is meaningless. Extreme values of the stimulus parameters could

result in different behavior than those for which participants were less sure, but this was not the case in the current work (models using a subset of data within one standard deviation of the standard parameter value showed the same significant effects and effect sizes).

Fitting the models for individual participants shows remarkable robustness and consistency of the effects of signal attributes on both intensity and duration judgments. The effects of signal duration and $f_o$ level were significantly positive for all participants in both tasks, and so were, naturally, the intensity effects in the intensity discrimination task. The effect of signal intensity on duration discrimination judgment was significantly positive on the group level (mixed effect model); on the individual level it was primarily manifested by consistency of effect direction (estimate sign) among the participants. In fact, this difference in statistical patterns between the intensity and duration discrimination tasks, and, in particular, between the $f_o$-level and intensity effects for duration discrimination provides further corroborative evidence to our fundamental claim that the stimulus frequency effect on duration perception is not solely mediated through the effect of fundamental frequency on perceived intensity of the sound.

A somewhat more complex situation arises with participants' sensitivity to frequency dynamics. The absolute dynamicity (the fact that one stimulus has greater range of $f_o$ contour than the other, regardless of the direction) had, at least for our stimuli, no significant and consistent effect on participants' judgments in either task. The $f_o$ slope direction also had no effect on intensity judgments. For durational judgments, however, the (more) rising stimuli were judged as longer than the (more) falling ones. Again, this effect was significant at the group level; presumably, this significance arose primarily through consistency among the participants in the direction of influence (individually, the effect was mostly non-significant). This result, including the relative lack of robustness, is generally consistent with the findings discussed in the Introduction: the evidence regarding this phenomenon is not unanimous and the results reported in literature suggest a degree of influence of participants' language background on the effect of stimulus dynamicity on durational judgments.

It is remarkable how well the participants were able to selectively attend to duration or intensity in making their task-specific judgments. This requires an ability to decompose the signal and can be quantified from the present measurements by comparing the relative influence of acoustic dimensions across tasks. Indeed, stimulus duration had 4.5 times stronger impact on duration judgments than loudness judgments (computed as the ratio of estimates in the statistical models). Similarly, the stimulus intensity had 4.7 times stronger impact on loudness judgments than on duration judgments. Hence, there is a symmetry in the relative impact of intensity and duration on perceived loudness and length.

The findings presented here must be taken into account by the energy integration models of the peripheral auditory system. The influence of fundamental frequency on perceived duration, and more importantly, the lack of interaction between frequency and intensity suggest an independent encoding of these signal characteristics, at least when used for durational judgments. In general, the observed pattern fits well with the dual klepsydra model for duration discrimination (*Wittmann, 2013*), assuming that the number of neural spikes entering the dual klepsydra system depends, in possibly interconnected but

cumulative way, on both the intensity and fundamental frequency of the sound. By design, the spectral energy was concentrated on a narrow area well separated from the fundamental frequency. The individual harmonics are then indiscernible and allow for the frequency band of the signal energy and the frequency of the pitch to be almost independent of each other.

In the current experiment, artificial stimuli were used with monolingual Finnish speakers, but the effect size coefficients indicate that these participants show effects in the same direction: increasing the intensity or fundamental frequency of a sound results in a perceived lengthening. Similar experimental data (of the duration discrimination task only) from speakers of other languages reported by *Šimko et al. (2015)* show qualitatively identical patterns for participants with different language backgrounds. Interestingly, the relative sensitivity to individual signal characteristics significantly differs for participants with different native languages, suggesting a degree of plasticity in auditory processing. Despite these quantitative differences, both results support the existence of general auditory biases and justify a generalization of the pattern of fundamental frequency influencing duration judgment independently of an intensity confound.

Previous research has questioned whether the subtle effects of these biases have any importance for human communication, namely, whether the well documented correlations between frequency patterns and duration of syllables, associated in many languages with phonological phenomena such as tone or quantity, arise primarily from production or perception constraints (*Gussenhoven & Zhou, 2013*; *Yu, 2010*). The findings reported here yield some support to the importance of taking the properties of auditory processing, even the subtle ones, into account when investigating phonetic characteristics of different languages and their origins. Moreover, the precise nature of these interplays partly determines the relevant, perceptually grounded degrees-of-freedom space in signal characteristics available for linguistic communication.

## CONCLUSIONS

Intensity, fundamental frequency, and duration of a sound influence its perceived intensity (loudness) and duration. The contributions of these signal characteristics to perceived duration and loudness are mutually independent and differ quantitatively between the two tasks of intensity and duration discrimination. At least for the material used in this study, fundamental frequency dynamics do not contribute consistently to either durational or loudness judgments. Fundamental frequency contribution to duration perception cannot be fully attributed to its effect on signal intensity, i.e, stimulus intensity alone cannot be responsible for duration judgment effects.

## ACKNOWLEDGEMENTS

Gratitude to Nora Fontell for helping in data collection.

# APPENDIX: DISCRIMINATION BY INDIVIDUAL SUBJECTS

The estimates and significances of duration difference, intensity difference, $f_o$ difference and dynamic range difference (in two versions) effects obtained by fitting (simple) logistic regression models for the data for each individual speaker separately. The results for intensity discrimination task are shown in Tables A1 and A2, those for the duration discrimination tasks in Tables A3 and A4. The models reported in Tables A1 and A3 use the difference in absolute values of dynamic $f_o$ range as a independent variable, those shown in Tables A2 and A4 use instead the alternative variable computed as a difference in raw range values.

**Table A1  Effect estimates and significances of the logistic models effects model fitted for the individual subjects for *intensity* discrimination (difference between absolute frequency ranges).** Significance codes: 0<*** < 0.001<** < 0.01<* < 0.05.

| Subject | Interc. | Dur. diff. | Int. diff. | Freq. diff. | Ran. diff. (abs) |
|---------|---------|------------|------------|-------------|------------------|
| 1 | 0.172 | 6.433*** | 0.199*** | −0.025 | −0.021 |
| 2 | 0.591*** | 4.866** | 0.201*** | 0.189*** | 0.067 |
| 3 | −0.307* | 4.907** | 0.481*** | 0.068* | 0.033 |
| 4 | 0.171 | 4.375*** | 0.153*** | 0.068* | 0.026 |
| 5 | 0.232 | 6.261** | 0.538*** | 0.284*** | 0.192*** |
| 6 | 0.319* | 8.787*** | 0.355*** | 0.233*** | −0.057 |
| 7 | 0.352* | 12.199*** | 0.270*** | 0.401*** | −0.022 |
| 8 | −0.187 | 4.372* | 0.661*** | 0.011 | 0.046 |
| 9 | −0.113 | 5.126*** | 0.163*** | 0.053* | −0.040 |
| 10 | 0.180 | 7.067*** | 0.204*** | 0.157*** | 0.052 |
| 11 | 0.174 | 8.685*** | 0.647*** | 0.128*** | 0.036 |

**Table A2  Effect estimates and significances of the logistic models effects model fitted for the individual subjects for *intensity* discrimination (difference between absolute frequency ranges).** Significance codes: 0<*** < 0.001<** < 0.01<* < 0.05.

| Subject | Interc. | Dur. diff. | Int. diff. | Freq. diff. | Ran. diff. (raw) |
|---------|---------|------------|------------|-------------|------------------|
| 1 | 0.180 | 6.487*** | 0.201*** | −0.030 | −0.034 |
| 2 | 0.594*** | 4.781** | 0.204*** | 0.190*** | −0.064* |
| 3 | −0.308*** | 5.006** | 0.480*** | 0.067* | 0.001 |
| 4 | 0.183 | 4.161** | 0.161*** | 0.070** | 0.064* |
| 5 | 0.258 | 6.245** | 0.505*** | 0.283*** | 0.006 |
| 6 | 0.300 | 9.354*** | 0.353*** | 0.240*** | −0.038 |
| 7 | 0.341* | 12.196*** | 0.271*** | 0.401*** | 0.016 |
| 8 | −0.203 | 4.336* | 0.662*** | 0.002 | −0.026 |
| 9 | −0.115 | 5.176*** | 0.166*** | 0.054* | 0.033 |
| 10 | 0.185 | 7.145*** | 0.209*** | 0.151*** | −0.030 |
| 11 | 0.185 | 8.705*** | 0.642*** | 0.129*** | 0.003 |

**Table A3  Effect estimates and significances of the logistic models effects model fitted for the individual subjects for *duration* discrimination (difference between absolute frequency ranges).**  Significance codes: 0<*** < 0.001<** < 0.01<* < 0.05.

| Subject | Interc. | Dur. diff. | Int. diff. | Freq. diff. | Ran. diff. (abs) |
|---------|---------|------------|------------|-------------|------------------|
| 1 | 2.141*** | 34.015*** | 0.043 | 0.338*** | 0.216** |
| 2 | 0.786*** | 24.618*** | 0.062 | 0.141*** | 0.002 |
| 3 | −0.225 | 42.528*** | 0.092 | 0.111** | 0.043 |
| 4 | −0.117 | 18.295*** | 0.044 | 0.084** | 0.067 |
| 5 | −0.140 | 47.741*** | 0.123* | 0.390*** | −0.045 |
| 6 | 0.525** | 27.480*** | 0.073 | 0.169*** | 0.046 |
| 7 | 1.147*** | 20.356*** | 0.100** | 0.368*** | 0.103 |
| 8 | 0.173 | 29.494*** | 0.068 | 0.150*** | −0.003 |
| 9 | 0.861*** | 18.146*** | 0.008 | 0.138*** | −0.024 |
| 10 | 0.618*** | 23.557*** | 0.035 | 0.169*** | −0.046 |
| 11 | −0.327 | 41.548*** | 0.189*** | 0.144*** | −0.024 |

**Table A4  Effect estimates and significances of the logistic models effects model fitted for the individual subjects for *duration* discrimination (difference between absolute frequency ranges).**  Significance codes: 0<*** < 0.001<** < 0.01<* < 0.05.

| Subject | Interc. | Dur. diff. | Int. diff. | Freq. diff. | Ran. diff. (raw) |
|---------|---------|------------|------------|-------------|------------------|
| 1 | 1.993*** | 32.261*** | 0.023 | 0.317*** | −0.002 |
| 2 | 0.773*** | 24.821*** | 0.062 | 0.140*** | −0.039 |
| 3 | −0.275 | 44.307*** | 0.085 | 0.126** | 0.119** |
| 4 | −0.094 | 18.270*** | 0.033 | 0.086** | 0.022 |
| 5 | −0.170 | 47.587*** | 0.124* | 0.392*** | 0.016 |
| 6 | 0.541** | 27.589*** | 0.077 | 0.169*** | 0.017 |
| 7 | 1.164*** | 20.255*** | 0.091* | 0.367*** | 0.016 |
| 8 | 0.129 | 29.669*** | 0.071 | 0.147*** | 0.055 |
| 9 | 0.859*** | 18.090*** | 0.010 | 0.135*** | 0.020 |
| 10 | 0.593*** | 23.935*** | 0.034 | 0.175*** | 0.054 |
| 11 | −0.328 | 43.165*** | 0.195*** | 0.156*** | 0.145** |

## Funding

This study was funded by the Academy of Finland (Grant numbers 1265610 and 1293348) and the Auditory Cognitive Neuroscience Erasmus Mundus Student Exchange Network. The funders had no role in study design, data collection and analysis, decision to publish, or preparation of the manuscript.

## Grant Disclosures

The following grant information was disclosed by the authors:
Academy of Finland: 1265610, 1293348.
Auditory Cognitive Neuroscience Erasmus Mundus Student Exchange Network.

## Competing Interests

The authors declare there are no competing interests.

## Author Contributions

- Caitlin Dawson analyzed the data, wrote the paper, prepared figures and/or tables, reviewed drafts of the paper.
- Daniel Aalto and Juraj Simko conceived and designed the experiments, analyzed the data, contributed reagents/materials/analysis tools, wrote the paper, prepared figures and/or tables, reviewed drafts of the paper.
- Martti Vainio conceived and designed the experiments, analyzed the data, contributed reagents/materials/analysis tools, wrote the paper, reviewed drafts of the paper.

## Human Ethics

The following information was supplied relating to ethical approvals (i.e., approving body and any reference numbers):

The University of Helsinki granted ethical approval to carry out the study within its facilities.

## Supplemental Information

Supplemental information for this article can be found online at http://dx.doi.org/10.7717/peerj.3734#supplemental-information.

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
