# Peer review of "The influence of fundamental frequency on perceived duration in spectrally comparable sounds"

_PeerJ, doi:10.7717/peerj.3734_

## Round 0.1 · original submission · Major Revisions

Further to your prior submission, the Reviewers found the manuscript much improved, but there are still significant issues to be addressed.

Pay particular attention to the comments relating to statistical power of the study, and the question about inter-subject variability which has been raised by both reviewers.

Also, representing intervals in semitones rather than in Hz may lead to ambiguity as to how the semitones were defined (even though one may presume equal temperament). Better to be explicit about the frequencies, as suggested by the reviewer.

Note that reviewers were confused about the use of gamma filtered sawtooth, as well as the use of the gliding stimuli, which must be better motivated.

Reviewer 1 ·

Basic reporting

In general, I think the authors have improved the manuscript significantly by removing the less relevant experiment and analyses. The language has become more attuned to the psychoacoustics literature, as had been commented on by Reviewer 2 in their first submission, although I still think the language needs to be more rigorous. For instance, I do not understand the statement "Physically, intensity of a (simple) sound is a function of its frequency" (lines 50-51). The use of the term "frequency" seems to be very non-specific towards the end, esp. in discussion.

The authors can better motivate their use of a gamma-toned filtered sawtooth (harmonic) stimuli, which I assumed is to isolate the effect of fundamental frequency (and its changes) from differences in spectral power.

Experimental design

no comment

Validity of the findings

It is unclear how the authors compare the relative impact of stimulus duration and intensity on duration & intensity judgement (lines 277-280).

I think it is important the author emphasize that this study shows the effect of fundamental frequency in isolation of spectral cue, but not to generalize (esp. in discussion) that as simply “frequency”. The fact that the stimuli in this study were gammatone filtered suggests that most of the harmonics will not be spectrally resolved in the ear. The authors briefly mentioned in discussion that their stimulus has very different spectral properties than previous studies (lines 266-268), but I would appreciate more elaboration, especially when relating to cited references. The Japanese study (Reference 30) cited by the authors to show difference in results used recorded (and manipulated) vowels as stimulus, and the authors ignored this fundamental difference while trying to attribute this to the language background of subjects.

How well do individual participants perform in the test? Is it possible to fit each participant independently? If so, do you see a difference in how intensity, duration and fundamental frequency relate to their judgement (e.g. direction of effect)? It was mentioned in line 291 that the effect size suggest "participants show effects in the same direction". Can the author explain how they see that?

Reviewer 2 ·

Basic reporting

Why is frequency gliding included in the study? No justification of this is given and it doesn’t seem relevant to the main study question. Was the direction of pitch change within frequency glides taken into consideration? Studies have shown that the direction of pitch change affects time perception. This could be part of the large variability in duration judgments for frequency glides observed.

Line 36: Might the neural spikes corresponding to the offset of a sound also be affected by the intensity and frequency of a sound?


Points for clarification:

Reference 4 on line 21 does not have to do with the state of the observer but the organization of consecutive sounds in time. This reference should be moved later in the sentence.

Lines 57-59: Can this point be clarified? I don’t understand what this means.

What are "multidimensional psychometric curves" (line 70)?

For the reader less familiar with the physical properties of sound, can you better explain why controlling the energy around the fundamental frequency spectral range resolves the confounding of intensity and frequency (lines 69-70).

Line 100: change duration to value

Please explain what nonlinear distortion tones are (line 131).

In figure 2: please explain what the dotted lines are indicating.

Line 156 says that it was hypothesized that “more dynamic f0 are perceived longer” but this is never stated in the introduction.

The phrase (line 159) “…the probability of a long sound was reduced by observing a very fast moving stimulus” is confusingly worded. Can this be rephrased?

Line 268: does “these stimuli” refer to the stimuli in the current study or previous studies?

Experimental design

I’m confused by the range of frequencies used (line 108). If an octave is made up of 12 semitones, how is 8 semitones above the mean of 150 Hz more than an octave above the mean (300 Hz)? I would recommend rerunning analyses using Hz instead of semitones.

Please explain why a gammatone filter (line 108) was used and what it means for an f0 to have unresolved harmonics (line 110).

Saying 600 ms of silence is confusing if white noise is playing through this period of “silence” (line 128).

What does it mean (lines 153-154) that the gliding speed is a source of bias but “requires perfect knowledge of the design which is not available to the participant directly…”?

Validity of the findings

The sample size in this study is noticeably small. Can the authors provide a power analysis to demonstrate that their study was sufficiently well powered to detect the effects they hypothesized?

The sentence starting with “In other words…” on line 224 is confusing. Can this point be clarified?

Lines 230-231, state that the simple regression equations of intensity and frequency predicting duration provide evidence that the impact of fundamental frequency on duration “is not entirely explained by the impact of intensity” but putting these variables into the model separately does not remove their shared variance from regression coefficients so these tests cannot support this assertion. Only the fact that when these two variables are included in the model together and they both still uniquely predict variance in duration judgments suggests that frequency has an influence on duration judgments beyond the influence of intensity.

Additional comments

This study fills an important gap in the literature on duration perception by disentangling the confound between pitch and intensity. As currently written there are a number of points that could be reworked for clarity. The power of the study should be addressed given the small sample size of the current study.

---

## Round 0.2 · accepted · Accept

Thank you for your work during the revision phase -- it considerably improved the clarity of the exposition and the scientific contextualization. I have not sent it again to the reviewers, but based on how you have addressed their earlier criticism, I decided to accept the current version as is.